Neutrophil extracellular traps in central nervous system (CNS) diseases

Shao Bo-Zong 1
Jiang Jing-Jing 2
Zhao Yi-Cheng 1
Zheng Xiao-Rui 1
Xi Na 1
Zhao Guan-Ren 1
Huang Xiao-Wu huangxw301@163.com 1
Wang Shu-Ling wangshuling0000@126.com wangshuling0000@123.com 3
1 Department of Pharmacy, Chinese PLA General Hospital , Beijing , China
2 Huazhong University of Science and Technology , Wuhan , China
3 Naval Medical University , Shanghai , China
Uversky Vladimir
Electronic publication date: 2024 Jan 4
Publication date: 2024
Volume: 12
Electronic Location ID: e16465
Received 2023 Jul 20; Accepted 2023 Oct 24
Copyright: ©2024 Shao et al.
Copyright year: 2024
Copyright holder: Shao et al.
License: This is an open access article distributed under the terms of the Creative Commons Attribution License, which permits unrestricted use, distribution, reproduction and adaptation in any medium and for any purpose provided that it is properly attributed. For attribution, the original author(s), title, publication source (PeerJ) and either DOI or URL of the article must be cited.
License URL: https://creativecommons.org/licenses/by/4.0/

Keywords: Neutrophil Extracellular Traps, Inflammation, Stroke, Neurodegenerative diseases, Cancer

Funding: The authors received no funding for this work.

==============================
Excessive induction of inflammatory and immune responses is widely considered as one of vital factors contributing to the pathogenesis and progression of central nervous system (CNS) diseases. Neutrophils are well-studied members of inflammatory and immune cell family, contributing to the innate and adaptive immunity. Neutrophil-released neutrophil extracellular traps (NETs) play an important role in the regulation of various kinds of diseases, including CNS diseases. In this review, current knowledge on the biological features of NETs will be introduced. In addition, the role of NETs in several popular and well-studied CNS diseases including cerebral stroke, Alzheimer’s disease, multiple sclerosis, amyotrophic lateral sclerosis (ALS), and neurological cancers will be described and discussed through the reviewing of previous related studies.

Introduction

Central nervous system (CNS) diseases refer to diseases that occur within the CNS, leading to the disturbance of structure and functions of CNS (Boziki et al., 2020; Kang et al., 2023; Nabors et al., 2020; Pellegrini et al., 2023). Despite the complexity of mechanisms underlying CNS diseases, recent studies have revealed that overactivation of inflammation and immunity is considered as a vital factor (Linnerbauer, Wheeler & Quintana, 2020; Rodriguez Murua, Farez & Quintana, 2022; Scheffer & Latini, 2020; Yang & Zhou, 2019). Duringinflammation- and immune-related responses, neutrophils are widely activated and play a significant role in triggering cascade inflammation and immune reactions(Kanashiro et al., 2020; Kara, Altunan & Unal, 2022; Khorooshi et al., 2020; Rossi, Constantin & Zenaro, 2020). In 2004, a special form of inflammatory reaction from neutrophils was initially reported and named neutrophil extracellular traps (NETs) (Brinkmann et al., 2004). NETs were characterized with the release of extracellular DNA traps. Such DNA traps were observed to consist of DNA fibers decorated with granule proteins. During the last two decades, along with the increasing knowledge of the physiological characteristics of NETs, the influences of NETs in various kinds of diseases were largely reported (de Jesus Gonzalez-Contreras & Zarate, 2022; Hidalgo et al., 2022; Khan et al., 2021; Papayannopoulos, 2018; Poto et al., 2022; Rada, 2019). However, so far, the role of NETs in CNS diseases is not fully elucidated, and the specific mechanisms remain unclarified. As a result, a study based on the reviewing of previous related literature is demanded to illustrate the whole picture on this issue. Our aim is to present a comprehensive view of NETs within the CNS system and offer novel insights into the exploration of mechanisms of CNS diseases for researchers in the field of CNS and biological science in this review. We believe that our article will offer novel insights into the investigation of CNS diseases.

Survey Methodology

PubMed (https://pubmed.ncbi.nlm.nih.gov/) and Web of Science (https://www.webofscience.com) were used to search for literature from 2000 to 2023. The search terms applied for literature reviewing included “neutrophil extracellular traps, central nervous system, cerebrovascular diseases, ischemic stroke, hemorrhagic stroke, neurodegenerative diseases, immune-related diseases, neurological cancer, infection, multiple sclerosis, amyotrophic lateral sclerosis, Alzheimer’s disease, inflammation, lipopolysaccharide, peptidyl arginine deiminase 4, inflammasome, autophagy, therapy”. The titles and abstracts of the relevant literature were initially screened, and the full texts were obtained.

Part I: Biological Features of Nets

So far, much knowledge has been obtained on the biological features of NETs. In the following contents, the general characteristics of NETs and its crosstalk between other inflammation- and immune-related mechanisms and processes including the inflammasome and autophagy will be described and discussed in detail (illustrated in Fig. 1).

Figure 1 Schematic illustration of crosstalk between NETs and inflammasome and autophagy.

NETs are highly associated with other inflammation- and immune-related mechanisms and processes including the inflammasome and autophagy. The pathways of the crosstalk between NETs and the inflammasome include the triggering of extracellular matrix remodeling process, cholesterol crystal challenge and induction of the TLR4-JNK pathway. For the crosstalk between NETs and autophagy, the possible pathways include the induction of the mTOR-dependent pathway, triggering neutrophil-mediated cell death process and mediation of platelet factor 4 from platelets.

General characteristics of NETs

Since its first reported in 2004, NETs were initially named with the term of “neutrophil extracellular trap-osis (NETosis)”. The term NETosis was widely applied on the basis of the reports showing that the extrusion of DNA stands could lead to neutrophil death, which contributed to the triggering of cascade inflammatory reaction (Berthelot et al., 2017; Carmona-Rivera & Kaplan, 2016; Desai et al., 2016; Jorgensen, Rayamajhi & Miao, 2017; Mahajan, Herrmann & Munoz, 2016). However, recent studies revealed that such release of DNA strands would not always lead to cellular death (Galluzzi et al., 2018; Hamam, Khan & Palaniyar, 2019; Papayannopoulos, 2018). As a result, the term “NETs” was recommended to replace “NETosis” as stated by the Nomenclature Committee on Cell Death (NCCD) in 2018 (Galluzzi et al., 2018). Hereafter, the term “NETs” or “NET formation” will be used instead of “NETosis” in the review.

According to modern knowledge on NETs, they were recognized as extracellular strands of decondensed (unwound) DNA fibers in combination of histones and neutrophil granule proteins including myeloperoxidase (MPO), matrix metalloproteinase (MMP), neutrophil elastase (NE), cathepsin G, complement factors, and other enzymatically active proteases and peptides (Demkow, 2021; Klopf et al., 2021; Linders et al., 2020; Varricchi et al., 2022). The triggering and active processes of NETs were previously summarized and described by us and other researchers (de Jesus Gonzalez-Contreras & Zarate, 2022; Hidalgo et al., 2022; Shao et al., 2021a). In general, under normal conditions, nuclei DNA strands in neutrophils are highly wrapped around histones into heterochromatin with protein-DNA interactions to constrain the potential energy for DNA extension and transcriptional activity (Shao et al., 2021a; Sorensen & Borregaard, 2016). However, in the challenge of certain stimuli such as exogenous invasion, intracellular calcium ion flux and agents including lipopolysaccharide (LPS) and phorbol 12-myristate 13-acetate (PMA), the condensed DNA strands in nucleus are uncoiled as fibrous polymers and released outside the nucleus to form NETs (Awasthi et al., 2023; Domer et al., 2021; Radermecker et al., 2019; Shao et al., 2021a). Based on the current knowledge, two proteases are vital in the process of NET formation, namely peptidyl arginine deiminase 4 (PAD4) and NE (Al-Kuraishy et al., 2022; Shao et al., 2021a; Zhu et al., 2022). PAD4 is in charge of catalyzing the conversion of arginine in histones to citrullines. The citrullination in histones leads to the weakening of the original positive charge of histones, failure of the strong histone-DNA binding and finally the decondensation of DNA strands. Although so far, NET formation independent of PAD4 has been reported (Kenny et al., 2017; Ravindran, Khan & Palaniyar, 2019), PAD4-dependent NETs are still commonly acknowledged as the classic form of NETs. The function of NE is to facilitate the destruction of histone-DNA binding via catalyzing and cleaving histones (Kumar et al., 2018; Wei et al., 2022). After decondensation, DNA strands decorated with histones and granule proteins are released extracellularly assisted by gasdermin D, a membrane protein functioning in the formation of pores (Shang et al., 2022; Vats et al., 2022). The release of NETs contributes to the further process in physiological and pathological conditions of organisms.

So far, a variety of signaling pathways have been reported involved in the process of NET formation. As previously demonstrated by us and others, many popular signaling proteins have been shown to regulate NET formation, including c-Jun N-terminal kinase (JNK), extracellular regulated protein kinase 1/2 (ERK1/2), Akt and Scr (Li et al., 2021b; Ma et al., 2022; Shao et al., 2021a; Small et al., 2022; Wright et al., 2020). In addition, phorbol 12-myristate-13-acetate (PMA), a protein kinase C (PKC) activator, has been widely used as an effective NET inducer in fundamental researches (An et al., 2019; Domer et al., 2021; Shirakawa et al., 2022). The involvement of various signaling proteins indicates the complexity of mechanisms processes for NET formation. Notably, the crosstalk between NETs and other vital inflammation- and immune related mechanisms, including inflammasome and autophagy, have been drawn increasing attention. Those contents will be discussed in detail in the following section.

Crosstalk between NETs and other inflammation- and immune-related mechanisms

Inflammasome, as a member of innate immunity, is a multi-protein oligomer activated under inflammation- and immune-related challenges (Bai et al., 2022; de Carvalho Ribeiro & Szabo 2022; Toldo et al., 2022). It is mainly produced in macrophages, neutrophiles, dendritic cells and other inflammatory and immune cells to contribute to the induction of inflammatory and immune responses via the recognition of pathogen-associated molecular patterns (PAMPs) or danger-associated molecular patterns (DAMPs) (Azumi et al., 2022; Biasizzo & Kopitar-Jerala, 2020; Herwald & Egesten, 2016). As previously reviewed by us, several kinds of inflammasomes have been reported, namely NOD-like receptor family, pyrin domain-containing (NLRP)1, NLRP2, NLRP3, NLR family caspase recruitment domain-containing protein 4 (NLRC4), and double-stranded DNA sensors absent in melanoma 2 (AIM2) (Shao, Cao & Liu, 2018; Shao et al., 2015). The inflammasome has been demonstrated to be involved in the pathogenesis and progression of various kinds of diseases in many systems, including cardiovascular diseases, autoimmune diseases, digestive system diseases, CNS diseases, respiratory diseases, metabolic diseases, malignant tumors and so on (Anderson et al., 2022; Andina, Bonadies & Allam, 2021; Jewell, Herath & Gordon, 2022; Qiu et al., 2022; Takahashi, 2022). So far, the crosstalk between the inflammasome and NETs has been widely reported in various kinds of diseases (Albrengues et al., 2018; Cristinziano et al., 2022; Hidalgo et al., 2022; Warnatsch et al., 2015). For instance, on the occurrence of atherosclerosis, NETs are triggered to release from neutrophils under the challenge of cholesterol crystals, which subsequently primes macrophages for inflammasome-related cytokine interleukin-1β (IL-1β) production and secretion (Warnatsch et al., 2015). In addition, NETs and NETs-mediated extracellular matrix remodeling contribute greatly to awaken the dormant lung cancer cells through LPS-mediated formation of inflammasomes in mice models (Albrengues et al., 2018). Although so far, little evidence is available to investigate the relations between NETs and inflammasome. However, it was recently reported that NETs could trigger the NLRP3 inflammasome activation and IL-18 secretion release during the occurrence of oxaliplatin-induced peripheral neuropathy (OIPN) via the induction of the LPS-toll-like receptor 4 (TLR4)-JNK pathway (Lin et al., 2022). However, to further explore the crosstalk between NETs and inflammasome, more studies are demanded for investigation.

Another mechanism worth mentioning is autophagy, which is commonly recognized as a vital metabolic mechanism to degrade and recycle long-lived proteins and damaged organelles (Kocak et al., 2022; Shao, Wang & Bai, 2022; Shao et al., 2021b). Since its initial discovery by Christian de Duve in the 1960s (Klionsky et al., 2016), autophagy has been increasingly studied and reported to participate in the pathogenesis and progression of various kinds of disorders (Ikeda, Zablocki & Sadoshima, 2022; Lei et al., 2022; Parmar et al., 2022). Autophagy is also highly involved in various inflammation- and immune-related diseases via the regulation of inflammatory and immune reaction, indicating autophagy as a vital mechanism of inflammation and immunity (Goswami, Karadarevic & Castano-Rodriguez, 2022; Hu et al., 2022b; Levine, Mizushima & Virgin, 2011; Tong et al., 2022). The crosstalk between NETs and autophagy has been increasingly studied in the recent decade (Guo et al., 2021; Huang et al., 2022b; Jiao et al., 2020; Skendros, Mitroulis & Ritis, 2018). Autophagy in neutrophils was proven to be highly associated with NETs-mediated cell death (Skendros, Mitroulis & Ritis, 2018). In addition, autophagy was reported to be involved in aging-related spontaneous formation of NETs via a mammalian target of rapamycin (mTOR) -dependent manner (Guo et al., 2021). On the occurrence of sepsis, neutrophil-mediated autophagy was shown to contribute to the systemic inflammatory responses and immune dysfunction via the triggering of autophagy-driven NET formation (Huang et al., 2022b; Jiao et al., 2020). In CNS diseases, Jin et al. (2022) reported that NET formation was mediated by platelet factor 4 (PF4) from platelet via the regulation of autophagy and contributed to thrombosis in patients with cerebral venous sinus thrombosis (CVST). However, because of the lack of evidence on the connection between NETs and autophagy, further studies are needed on this issue.

Part II: NETs in CNS Diseases

In the recent two decades, the role of NETs in CNS diseases and its underlying mechanisms have been increasingly investigated. In the following contents, NETs in several kinds of popular and well-studied kinds of CNS diseases including cerebral stroke, Alzheimer’s disease, multiple sclerosis, amyotrophic lateral sclerosis (ALS), and neurological cancers will be described and discussed through the reviewing of previous related studies (illustrated in Fig. 2).

Figure 2 Schematic illustration of role of NETs in CNS diseases.

NETs released from neutrophils may produce effects on the pathogenesis and progression of cerebral stroke via promoting thrombosis, influencing cerebral revascularization and vascular remodeling, promoting cerebral hemorrhage, and aggravating cerebral tissue damage. In Alzheimer’s disease, NETs may regulate complement system activation, DNase-related genetic mutation, and LFA-1 integrin mediation. In multiple sclerosis, NETs may produce a regulatory effect through changing IgG and IgG-IgM complex and regulating CD4+ T cells and lymph nodes connection. In amyotrophic lateral sclerosis, NETs may lead to the neurodegenerative process and peripheral motor pathway damage. In neurological cancers, NETs may regulate glioma and tumor microenvironment connection, enhance hypercoagulability and damage brain-blood barrier or brain-tumor barrier.

NETs in cerebral stroke

Cerebral stroke, also known as cerebrovascular accident, is a group of acute cerebrovascular diseases (Hackam & Hegele, 2022). It leads to the damage to brain tissue caused by a sudden rupture of a blood vessel in the brain or a blockage that prevents blood from flowing to the brain (Minhas et al., 2022; Pan et al., 2022; Tuo, Zhang & Lei, 2022). Cerebral stroke includes ischemic stroke and hemorrhagic stroke, and leads to large amount of death with high incidence and teratogenic rate (Grysiewicz, Thomas & Pandey, 2008). On the occurrence of cerebral stroke, especially ischemic stroke, cerebrovascular thrombosis serves as a vital factor (Ding et al., 2022; Hu et al., 2022a; Zhou et al., 2022). So far, NETs have been widely reported to be important in the pathological process of cerebral stoke, and the role of NETs in stroke is most studied among all kinds of CNS diseases. The level of NET formation was detected to be high in both venous and arterial thrombosis, and NET-specific citrullinated histone 3 (citH3) was proven to possibly constitute a useful prognostic marker and target in patients with acute stroke (Laridan, Martinod & De Meyer, 2019). In the process of thrombosis, NETs were reported to contribute greatly to the formation of arterial or venous thrombi (Laridan, Martinod & De Meyer, 2019). The formation of NETs was shown to promoted by activated platelets in stroke and could, in turn, activate platelets, thus favoring thrombotic processes (Laridan et al., 2017; Laridan, Martinod & De Meyer, 2019). In addition, NET-related variations were observed in thrombus structure in patients with ischemic stroke as well as other vascular diseases including coronary disease and peripheral artery disease. Such variations were shown to produce a significant influence on thrombus stability (Farkas et al., 2019).

NETs were also shown to influence cerebral revascularization and vascular remodeling after stoke (Abbasi et al., 2022; Kang et al., 2020; Mohamud Yusuf et al., 2021). It was reported that disruption of NET formation by DNase 1 as well as genetical or pharmacologic inhibition of PAD4, one of the important enzymes for NET formation mentioned above, increased neovascularization and vascular repair during stoke recovery (Kang et al., 2020). In addition, increased NETs, as well as fibrin and von Willebrand Factor (vWF) composition in thrombi, might reduce the likelihood of revascularization by altering thrombus mechanical properties in patients with stroke (Abbasi et al., 2022).

Cerebral hemorrhage is another vital factor for stroke besides cerebral ischemia. NETs were also demonstrated to influence the pathological processes of cerebral hemorrhage. Wang et al. (2021) reported that NETs could promote tPA-induced cerebral tissue plasminogen activator (tPA) via downregulation of cyclic GMP-AMP synthase (cGAS) in stroke patients. In addition, the application of RNase was shown to suppress NET formation in mice subarachnoid hemorrhage models (Fruh et al., 2021). However, despite of much evidence on the role and mechanisms of NETs in cerebral stroke in clinical and animal studies, yet seldom related agents or therapies are available to be successfully applied in clinical practice. Further studies are demanded for investigation.

NETs in Alzheimer’s disease

Alzheimer’s disease belongs to neurodegenerative diseases. Neurodegenerative diseases are a group of diseases characterized by progressive loss of neuronal structure and function, with the result of neuronal death (Haass & Selkoe, 2007). So far, three neurodegenerative diseases are well-studied and popular, including Alzheimer’s disease, Parkinson’s disease and Huntington’s disease (Fang et al., 2020; Peng, Trojanowski & Lee, 2020; Ross & Tabrizi, 2011). According to current knowledge on neurodegenerative diseases, abnormal protein aggregation in neurons serve as the major cause of those three diseases, including Alzheimer’s disease with β-amyloid accumulation, Parkinson’s disease with α-synulein aggregation to form proteinaceous cytoplasmic Lewy bodies and Huntington’s disease with aggregate-prone huntingtin protein accumulation (Frisoni et al., 2022; Liu et al., 2015; Shao, Cao & Liu, 2018; Zhang et al., 2019). Several factors are considered as possible factors for the pathogenesis of neurodegenerative diseases, including aging, heredity, geography, overwhelming inflammatory and immune reaction and so on (Bloem, Okun & Klein, 2021; Jessen et al., 2020; Leng et al., 2019; Scheltens et al., 2021).

The role of NETs in Alzheimer’s disease has been drawn increasing attention in the last decade. As previously stated, β-amyloid aggregation in neurons led to increasing production and activation of complement system components including C1q, CR1 and C5a, which subsequently assembled neutrophils to the brain and induced NET extrusion (Kretzschmar et al., 2021). The process of NETs deposition further triggered complement system activation through the alternative pathway and properdin binding (Kretzschmar et al., 2021; Yuen et al., 2016). The inability of complement system for NETs degradation led to the accumulation of NETs and overactivation of complement system, thus resulting in the cascade inflammatory and immune responses and damage to neurons (de Bont, Boelens & Pruijn, 2019; Kretzschmar et al., 2021). In addition, DNase, a commonly used NETs inhibitor, has been successfully applied in the treatment of Alzheimer’s disease, and genetic variants in DNase genes including DNASE1, DNASE2 and DNASE1L3 can result in the downregulation of DNase expression in Alzheimer’s disease (Kretzschmar et al., 2021; Tetz & Tetz, 2016). Another study showed that NET formation contributed to Alzheimer’s disease-like pathogenesis and cognitive impairment via the mediation of LFA-1 integrin (Zenaro et al., 2015). These studies indicate that NETs play an important role in the pathogenesis and progression of Alzheimer’s disease. Although so far, little evidence was available for the connection between NETs and Parkinson’s disease and Huntington’s disease, we believe that regulating NETs might serve as a breakthrough in the treatment of those two diseases, and might be an attractive research direction for future studies.

NETs in multiple sclerosis

Multiple sclerosis is a commonly diagnosed autoimmune disease, characterized as inflammatory demyelination of the white matter of CNS (Carotenuto et al., 2022; Correale, Hohlfeld & Baranzini, 2022; Pitt et al., 2022). Its prevalence is estimated at approximately 2.5 million people worldwide, with 70% patients between 20–50 years (De Bondt et al., 2020; Thompson et al., 2018). Although the specific pathogenesis of multiple sclerosis has not been clarified, several factors are recognized as the possible causes, including genetic variants, environmental factors, gender, age, smoking, infection especially in the CNS, vitamin D deficiency and so on (Cortese et al., 2020; Leray et al., 2016; Oturai et al., 2021).

Among all factors for the pathogenesis of multiple sclerosis, the state of inflammatory and immune dysregulation is considered as a vital factor (Perez-Jeldres, Alvarez-Lobos & Rivera-Nieves, 2021; Rodriguez Murua, Farez & Quintana, 2022). In this process, the activation of neutrophils and the subsequent release of NETs are one of the important components for such dysregulation state (Costanza et al., 2019; De Bondt et al., 2020; Maatta et al., 1998; Naegele et al., 2012; Paryzhak et al., 2018). It has been reported that the number of neutrophils is increased in both the periphery and CNS before and during the occurrence of multiple sclerosis (Maatta et al., 1998). NETs produced and released by activated neutrophils has been revealed to be enhanced in the serum of multiple sclerosis patients compared with healthy people (Naegele et al., 2012). Additionally, NET-related proteases were shown to alter the glycan composition on circulating IgG molecules and IgG-IgM immune complexes in multiple sclerosis, potentially influencing the severity of multiple sclerosis (Paryzhak et al., 2018). The DNA extrusions were demonstrated to convey autocrine costimulatory signals to CD4+ T cells and lymph nodes during the priming phase of multiple sclerosis in mice models (Costanza et al., 2019). These results indicate the effect of NETs on multiple sclerosis. However, so far, seldom therapies are available for the successful application in the treatment of multiple sclerosis in clinical practice. We believe that with the increasing study on this issue, therapeutic strategy regulating NETs will be developed in the future.

NETs in ALS

ALS, also known as motor neuron disorder, is a fatal CNS neurodegenerative disease characterized by the degeneration of both upper and lower motor neurons (Feldman et al., 2022; Hardiman et al., 2017). The etiology of ALS remains unclarified. Genetic mutations and hereditary factors are recognized to be highly related to the pathogenesis and progression of ALS (Johnson et al., 2021). Besides, environmental factors such as heavy metal pollution may also serve as risk factors (Li et al., 2021a).

Despite the complication of mechanisms for ALS, several pathological events for the onset and development of ALS have been reported, including excessive glutamate levels, overactivation of inflammatory and immune responses, protein misfolding, blood–brain barrier damage, oxidation reaction product accumulation, and reduced energy metabolism (Cao & Fan, 2023). Those pathological factors may contribute to the neurodegenerative process of ALS (Cao & Fan, 2023). As previous reported, neutrophils are closely associated with ALS (Murdock et al., 2021). As an important component of neutrophil-mediated inflammatory reaction, the role of NETs in ALS is increasingly studied by researchers. It was previously reported that NET release could lead to the process of peripheral motor pathway damage in an SOD1G93A rat ALS model (Trias et al., 2018). The administration of the tyrosine kinase inhibitor drug masitinib to suppress neutrophil and mast cell activation and infiltration significantly alleviated the severity of ALS. The study on the effects of NETs on ALS may provide a potential pathway for the exploration of mechanisms for ALS and development of novel therapeutic strategies for the treatment of ALS.

NETs in neurological cancers

Neurological cancers are a group of primary or metastatic cancer of the CNS (Balachandran et al., 2020; Shao et al., 2021a). According to a report from the World Health Organization (WHO) in 2016, neurological cancers include diffuse astrocytic and oligodendroglial tumors, other astrocytic tumors, ependymal tumors, other gliomas, lymphomas of the CNS, and metastatic cancers (Louis et al., 2016; Shao et al., 2021a). It is widely acknowledged that the tumor-related inflammatory and immune microenvironment plays a significant role in the pathogenesis and progression of various malignant tumors. This suggests that the regulating local and overall inflammatory and immune responses could be an effective and promising strategy for cancer treatment(Barkley et al., 2022; Hiam-Galvez, Allen & Spitzer, 2021; Peng et al., 2022). It was previously reviewed by us that NETs played an important role in several kinds of cancers, including breast, lung, colorectal, pancreatic, blood, neurological and cutaneous cancers, mainly through the regulation of inflammatory and immune state and related signals and mechanisms (Shao et al., 2021a).

In neurological cancers, it has beenreported that NETs generate by tumor-infiltrating neutrophils (TINs) regulated the connection between glioma and tumor microenvironment in patients with malignant glioma via the mediation of HMGB1/RAGE/IL-8 axis (Zha et al., 2020). Moreover, NET formation has been shown to increase the hypercoagulability in glioma patients, suggesting that targeting NETs might be an effective approach for preventing thrombotic complication in glioma patients (Zhang et al., 2021). In addition, NETs were reported to damage the brain-blood barrier or brain-tumor barrier, which facilitated the development and metastases of glioma (Lin et al., 2021). However, to take advantage of NETs regulation in the treatment of neurological cancers, further studies are demanded.

Therapeutic applications of NETs in the treatment of CNS diseases

As described and discussed above, NETs play a significant role in alleviatingCNS diseases including cerebral stroke, Alzheimer’s disease, multiple sclerosis, ALS, and neurological cancers. So far, an increasing number of studies have reported several therapeutic strategies against those diseases targeting NETs.

In cerebral stroke, several therapies taking advantage of NET suppression have been widely reported. As revealed by Dhanesha et al. (2022), nuclear pyruvate kinase muscle 2 (PKM2), a modulator of systemic inflammation, was shown to enhance neutrophil activation on the occurrence of ischemic stroke. The administration of PKM2 nuclear translocation inhibitors could significantly alleviate neutrophil hyperactivation and neutrophil-mediated NET release, thus improving the short-term and long-term functional outcomes after stroke. Another previous study demonstrated that Edaravone Dexborneol, which was comprised of two active ingredients including Edaravone and (+)-Borneol, exerted an alleviative effect on acute ischemic stroke (Huang et al., 2022a). Such effect was mediated by the amelioration of NET-induced blood–brain barrier damage. In addition, the intravenous application of RNase A (the bovine equivalent to human RNase 1) was reported to alleviate subarachnoid hemorrhage through the abrogation of NET burden in the brain parenchyma (Fruh et al., 2021). Such findings indicated the potential therapeutic value of NET inhibition in cerebral stroke.

In Alzheimer’s disease, Serebrovska et al. (2019) showed that intermittent hypoxia-hyperoxia training could improve the cognitive functions in pre-Alzheimer’s disease patients and slow down the development of Alzheimer’s disease through the suppression of NET-mediated blood–brain barrier damage and brain parenchyma destroy. In addition, it was revealed that dimethylfumarate, a commonly used food additive, could alleviate neutrophil-mediated chronic inflammatory diseases including multiple sclerosis through the inhibition of neutrophil activation and suppression of NET formation (Muller et al., 2016). Those studies exerted that NETs might serve as potential targets in the treatment of such CNS diseases.

Conclusions

Taken together, in our current study, we introduced the biological characteristics of NETs and crosstalk between other inflammation- and immune-related mechanisms. In addition, we described and discussed the roles and potential mechanisms of NETs in serval well-studied CNS diseases including cerebral stroke, Alzheimer’s disease, multiple sclerosis, ALS, and neurological cancers, through a review of previous related studies. We believe that our study will provide novel insight into the exploration of CNS diseases in terms of pathogenesis and progression and treatment.

Additional Information and Declarations

Competing Interests

Author Contributions

Data Availability

The authors declare there are no competing interests.

Bo-Zong Shao conceived and designed the experiments, performed the experiments, prepared figures and/or tables, authored or reviewed drafts of the article, and approved the final draft.

Jing-Jing Jiang conceived and designed the experiments, authored or reviewed drafts of the article, and approved the final draft.

Yi-Cheng Zhao conceived and designed the experiments, authored or reviewed drafts of the article, and approved the final draft.

Xiao-Rui Zheng performed the experiments, prepared figures and/or tables, and approved the final draft.

Na Xi conceived and designed the experiments, authored or reviewed drafts of the article, and approved the final draft.

Guan-Ren Zhao analyzed the data, authored or reviewed drafts of the article, and approved the final draft.

Xiao-Wu Huang analyzed the data, authored or reviewed drafts of the article, and approved the final draft.

Shu-Ling Wang performed the experiments, prepared figures and/or tables, and approved the final draft.

The following information was supplied regarding data availability:

This is a literature review.

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
