# Peer review of "Neutrophil extracellular traps in central nervous system (CNS) diseases"

_PeerJ, doi:10.7717/peerj.16465_

## Round 0.1 · original submission · Major Revisions

Please address the concerns of all reviewers and amend the manuscript accordingly.

**Language Note:** The review process has identified that the English language must be improved. PeerJ can provide language editing services - please contact us at [email protected] for pricing (be sure to provide your manuscript number and title). Alternatively, you should make your own arrangements to improve the language quality and provide details in your response letter. – PeerJ Staff

Reviewer 1 ·

Basic reporting

The review is well written and comprehensive.

The figure is not up to the mark. While the flow looks good, the fonts are extremely small. Also, coloring the boxes with different types of outline colors does not do justice to a well written review article.
Suggestion: Make fonts bigger; remove bright colors or just keep B/W boxes that have separate bullets for each topic.

Experimental design

Gaps in the literature are correctly identified and addressed. Nice flow and logic of the review. This is an important topic within the scope oF CNS diseases in general and is focused to the scope of PeerJ.
Appropriate references are provided as required.

Validity of the findings

As a review article the existing literature is properly contextualized. It reinforces the role of Neutrophils and NETS in CNS disease, especially neurodegenerative diseases.

Additional comments

Authors must include the role of NETS in ALS. Nowhere this function is discussed or mentioned.
Two references must be discussed in this context:
1) Trias E, King PH, Si Y, Kwon Y, Varela V, Ibarburu S, et al. Mast cells and neutrophils mediate peripheral motor pathway degeneration in ALS. JCI Insight (2018) 3(19). doi: 10.1172/jci.insight.123249

2) Front. Immunol., 16 August 2023
Sec. Multiple Sclerosis and Neuroimmunology
Volume 14 - 2023 | https://doi.org/10.3389/fimmu.2023.1246768

Major Comments: 1. With the modification of figure as suggested and 2. With incorporation of discussion on the above two references and the role of NETS in ALS, the review can be accepted.

Reviewer 2 ·

Basic reporting

In this manuscript, Shao et al. discuss Neutrophil extracellular traps (NETs) and summarize the role of NETs in immune related mechanisms and in CNS diseases including cerebral strokes, Alzheimer's Disease, Multiple Sclerosis and neurological cancers. There are several issues that need to be addressed:

1. The English language should be improved and the manuscript needs a thorough grammar check. I suggest that the authors to contact a professional editing service.

2. The introduction is repetitive. NETs should be introduced and discussed more in detail. The authors should highlight the existing knowledge gap and how this review serves to fill those gaps.

3. In the introduction, the authors mention "In CNS diseases, the role of NETs has been largely revealed and reported [24-26]. However, so far, the role of NETs in CNS diseases is not fully elucidated, and the specific mechanisms remain unclarified." Please rephrase this as these sentences are contrary to each other.

4. The authors should include a diagram to show all the factors that contribute to NET release.

5. It will be informative to include a diagram to show how NETs crosstalk with the inflammation/immune pathways

6. Since the authors majorly discuss NETs in AD, I suggest the authors to change the section heading from "NETs in neurodegenerative diseases" to "NETs in Alzheimer's Disease".

7. The authors should include a section to discuss the therapeutic application of NETs in detail.

Experimental design

N/A

Validity of the findings

N/A

·

Basic reporting

In this review the authors detailed the current understanding of role of NETs in CNS diseases.

I ask the authors to address the following comments.

The section describing survey methodology is misleading. This review is reading like an expert review. But this section is written as it is a systematic review. I suggest authors to remove this section (lines 76-84).

Sentences in lines 65-67 are contradictory to each other.

Line 67: I suggest the authors not use the term ‘study’ as it is a review article.

Authors used the phrase ‘important disease’ to indicate commonly occurring/well studied diseases. Please note that the word ‘important’ is not making sense.

Experimental design

-

Validity of the findings

-

---

## Round 0.2 · accepted · Accept

All issues pointed out by the reviewers were adequately addressed and I am glad to recommend your revised manuscript for publication.

Reviewer 2 ·

Basic reporting

The authors have successfully addressed all of my comments. The manuscript is now acceptable for publication in PeerJ

Experimental design

N/A

Validity of the findings

N/A

Additional comments

N/A

·

Basic reporting

The authors addressed my concerns. I have no further comments.

Experimental design

-

Validity of the findings

-